# Ultrasonic flexural mode emitters: New approaches to increasing sound pressure during oscillation generation in gas media

Vladimir Khmelev, Andrey Shalunov ⓘ*, Sergey Tsyganok, Alexander Sinkin, Pavel Danilov ⓘ

Biysk Technological Institute (branch) Altai State Technical University, Biysk, Russia

* shalunov@u-sonic.ru

## Abstract

This work is devoted to the development and study of ultrasonic flexural-oscillating disk emitters for gas environments, generating elastic vibrations at ultrasonic frequencies (above 20 kHz) with high sound pressure levels required for energy-intensive technological processes (sound pressure levels exceeding 140 dB). The aim of the study was to identify the limitations of traditional flat disk designs and to substantiate new technical solutions that can significantly improve radiation efficiency in gas environments. The study demonstrated that the use of a flat titanium disk with a diameter of 146 mm, operating in the second bending mode, yields a sound pressure level of 147.5 dB (471 Pa), while the implementation of a stepped-profile surface of the same diameter increases the SPL to 153.2 dB (914 Pa). The subsequent use of phase-aligning horns and rear reflectors enabled a record-breaking sound pressure level of 159.2 dB (1824 Pa), more than double the original values and significantly exceeding the performance of known analogues. For comparison, speakers with stepped-profile disks with diameters of 250, 320, 360, and 410 mm, operating in the third, fourth, and higher vibration modes, were manufactured and tested. Despite the increase in radiating surface area, the achieved sound pressure levels were 140 dB (195 Pa), 143 dB (350 Pa), 148 dB (435 Pa), and 150 dB (700 Pa), respectively, which turned out to be lower than those of emitters with a diameter of 146 mm, operating in the second mode. The results confirm the feasibility of creating emitters operating specifically in the second flexural mode and demonstrate the advantages of stepped-profile disk geometry and phase-equalizing horn systems. The resulting solutions enable the generation of directional acoustic fields with pressure levels exceeding at minimum 155 dB and at maximum 159 dB. In turn, these techniques open up broad application prospects in air purification, aerosol precipitation, fire safety, and industrial drying and defoaming processes.

**Data availability statement:** All relevant data are within the manuscript and its Supporting information files.

**Funding:** The author(s) received no specific funding for this work.

**Competing interests:** The authors have declared that no competing interests exist.

## 1. Introduction

The formation of elastic oscillations of ultrasonic (US) frequency (higher than 22 kHz) in gas medium poses a significant practical interest. Such oscillations allow information to be transmitted over long distances in a range inaccessible to human perception, including in difficult weather conditions, i.e., fog, rain, snow, smoke, as well as in conditions of acoustic and light interference [1–3]. Ultrasonic emitters for creating such oscillations are used in security systems [4], for repelling animals [5] and for suppressing sound recording devices [6].

Additionally, ultrasonic vibrations have a number of useful effects: they help clean the air from solid and liquid particles, settle smoke during fires [7–9], allow for low-temperature drying of food, medicines and explosives [10–12]. They remove liquids from surfaces [13] and are used for defoaming in chemical processes [14,15] and to apply coatings and other technological operations [16]. A sharp increase in the efficiency of such processes (coagulation, drying, defoaming, etc.) is achieved when exposed to ultrasonic vibrations with a SPL (sound pressure level) of at least 135–145 dB [17]. This, in turn, requires the creation of sources of ultrasonic vibrations capable of generating high levels of acoustic energy in gas environments.

The first attempts to generate ultrasound in gas environments date back to the first half of the 20th century. The most famous are the Hartman gas-jet emitters and their modifications [18,19]. The principle of their operation was that a jet of gas flowing out of a nozzle excited vibrations in the resonator, forming an acoustic field in the ultrasonic frequency range. However, the efficiency of such devices remained low. The efficiency did not exceed 15–20%, and the power of the emitted vibrations rarely exceeded several watts at a SPL of up to 120–130 dB [19]. While this was sufficient for a number of experimental studies and simple tasks (such as repelling animals or acoustic signaling), these parameters were insufficient for technological processes requiring intense sonication (defoaming, particle coagulation, accelerated drying) [19].

Furthermore, gas-jet systems were bulky, dependent on a stable gas flow, and had poor spectral stability [20]. These limitations stimulated the search for alternative solutions that would generate a powerful and more controllable ultrasonic field in air and other gaseous media.

A significant breakthrough was achieved with the advent of piezoelectric transducers. The work of P. Langevin in the early 20th century, followed by the research of Mason, Redwood, and others, laid the foundation for the development of high-power ultrasonic systems.

A classic Langevin transducer is a "sandwich" structure: piezoelectric elements are clamped between metal masses and excited by an electrical signal, creating longitudinal mechanical vibrations. These vibrations are amplified and transmitted to the working medium using concentrators and horns.

Transducers of this type have found wide application in industry (plastic and metal welding, ultrasonic cleaning, medical devices), as well as in scientific research. However, their efficiency in gaseous environments is significantly lower than in liquids or solids. This is due to the significant difference in acoustic impedance between the transducer material and air.

The development of Langevin transducer theory was accompanied by the extensive use of equivalent electrical circuits, one-dimensional models, and numerical methods. The work of Iula et al., Abdullah, and Pak demonstrated the potential of three-dimensional FEM modeling for analyzing vibration modes and frequency responses. Wang and Tsai proposed block diagram methods for finding resonant frequencies. These studies made an important contribution to understanding the behavior of transducers and improved design accuracy.

However, despite advances in modeling and design improvements, Langevin transducers have not proven to be the optimal solution for airborne operation. Their effectiveness in gaseous environments remains limited, and the sound pressure levels they generate rarely exceed 115–125 dB. This necessitated the search for specialized designs specifically designed for airborne operation.

The most promising direction for the development of solid-state ultrasound systems is emitters based on flexural vibrations of disks. Their fundamental advantage lies in the better matching of the wave impedance between the metal and the gas medium. Unlike longitudinal waves, flexural waves propagate along the disk at a slower speed, facilitating the transfer of energy into the air.

Furthermore, the disks have a large radiating surface, enabling them to generate significant sound pressure levels in a relatively compact package. This makes them suitable for creating powerful ultrasonic systems used in air purification, aerosol separation, fire safety (smoke separation), and industrial processes.

Studies have shown that small-diameter disks (less than 50 mm) excited by the fundamental oscillation mode are capable of operating at frequencies above 20 kHz. However, their radiating surface is too small, and their radiation pattern is too broad. To generate more powerful ultrasonic fields, larger-diameter disks (100–400 mm) excited by higher oscillation modes have been used.

A key problem arises here. When the second and subsequent modes are excited, zones oscillating in antiphase are formed on the disk surface. These zones partially cancel each other out, reducing the resulting sound pressure and the system's efficiency. Experiments have shown that as the disk diameter increases, the proportion of the surface operating in antiphase increases, limiting the capabilities of simple flat designs.

Several solutions have been proposed to overcome these limitations. The first is the use of a stepped disk profile, with the thickness varying radially. This design compensates for phase shifts between zones and achieves more consistent radiation. Research by V. Khmelev, A. Shalunov, and their colleagues demonstrated that a stepped disk with a diameter of 146 mm increases the SPL from 147.5 dB (flat disk) to 153.2 dB, nearly doubling the acoustic power.

Later, various varieties of these emitters were created, such as: focusing emitters [21], rectangular stepped plate emitters [22,23], stepped plate emitters combined into gratings [24] and a two-frequency stepped plate emitter [25] designed for an acoustic parametric array source [26].

Another approach has been the use of horns and reflectors. Andrés et al. developed a system with a rectangular plate and reflectors, which provided coherent radiation and an efficiency increase of up to 70% [27,28]. Reflectors allow the energy of both surfaces of the disk to be utilized and the phases of antiphase zones to be matched, similar to the principle of Fresnel lenses.

Thus, the review shows that:

1. gas-jet emitters have been used for a long time, but were limited by low efficiency;

2. Langevin transducers perform well in liquids and solids, but are not optimal for gaseous media;

3. disk emitters offer the greatest potential for generating powerful ultrasonic fields in air; however, the problem of the difference in the wave impedances of the emitter and air and the need to ensure consistent in-phase radiation into the gas from disk sections oscillating in antiphase remains relevant.

Therefore, this article is devoted to improving the efficiency and expanding the functionality of ultrasonic emitters for gaseous media. The following tasks must be addressed:

determine the conditions for generating ultrasonic vibrations with the highest possible sound pressure level using disk emitters;

- Select design solutions that provide a given area and uniform energy input into the gaseous medium;

- develop new approaches to the formation of directed radiation capable of providing the required effect either in a specific area of space or at a certain distance from the source.

## 2. Methodology

### 2.1. Methods for constructing disk emitters

As noted earlier, the key element of ultrasonic emitters designed for operation in gas environments are metal disks that perform bending vibrations. This is due to the fact that the wave resistance of a bending vibrating metal disk is better matched to the wave resistance of the gas, compared to emitters of longitudinal vibrations. Such matching is achieved due to the fact that the propagation speed of bending vibrations is approximately two times lower than the speed of longitudinal waves in metal [17,28].

In practice, it is necessary to use disk emitters of different diameters. However, effective excitation of oscillations is possible only if the excitation frequency coincides with the natural resonant frequencies of the disk. This is the main technical difficulty: with the characteristic speed of propagation of bending oscillations in a titanium disk (about 2000 m/s), it is possible to achieve a frequency of at least 20 kHz only with a disk diameter of less than 50 mm. This corresponds to an area of the radiating surface of about 20 cm².

It is obvious that a radiator of such a small size forms a wide directional pattern (more than 45°) and provides a low level of acoustic power (less than 1 W). In this regard, to generate ultrasonic vibrations in a gas environment with a frequency of over 20 kHz and a SPL of over 135–140 dB, radiators are used that are excited in the second and subsequent modes of disk vibrations, while the area of their radiating surface exceeds 100 cm².

Thus, a logical solution to the problem of increasing the radiation efficiency is to use disks of a larger diameter, excited at higher oscillation modes (second, third, and so on). This allows to increase the output of acoustic energy radiated into the gas medium proportionally to the increase in the disk area.

To confirm or refute the effectiveness of this approach, it is necessary to develop and study the designs of ultrasonic emitters with an operating frequency of more than 20 kHz, equipped with disks with a diameter of 100–300–400 mm (radiating surface up to 1000 cm²). In this case, the disks should operate at the 2nd, 5th, 6th, and even 7th oscillation modes.

Therefore, the materials presented below will be aimed at a comparative analysis of known and new technical solutions for ultrasonic emitters in order to identify the most effective designs that provide a high energy output into the gas environment.

The principle of constructing such emitters is illustrated in Figs 1 and 2. They show the design diagram of the device and the vibration patterns of a flat metal disk mechanically and acoustically connected to a piezoelectric transducer [28–30].

As shown in Fig 2, the disk can oscillate in one of the resonant modes, each of which is characterized by a certain distribution of amplitudes, including the presence of zones that provide the emission of oscillations into the gas with different phases, nodal lines in which the oscillation amplitude is zero.

The shown design of the emitter allows us to understand the principles of formation of bending oscillations, however, it has not received practical application due to the uneven distribution of amplitudes over the disk surface. In such a disk, the oscillation amplitude decreases with distance from the center, which reduces the radiation efficiency of peripheral ring zones 2 and 3 (Fig 3).

The decrease in the oscillation amplitudes is explained by the increase in the rigidity of the disk's ring zones as they move away from the center and their area increases. The exception is the outermost ring, the rigidity of which is lower due

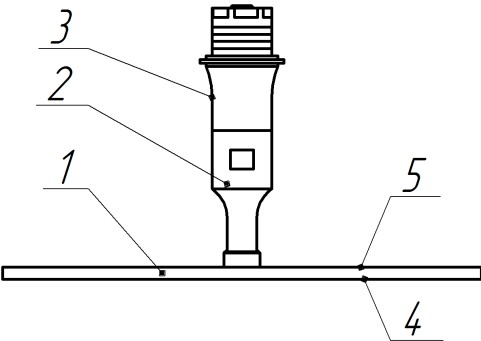

**Fig 1. Flat disk emitter.** 1 – flexurally oscillating disk; 2 – ultrasonic oscillations concentrator; 3 – piezotransducer; 4 – emitting side of the disk; 5 – disk backside.

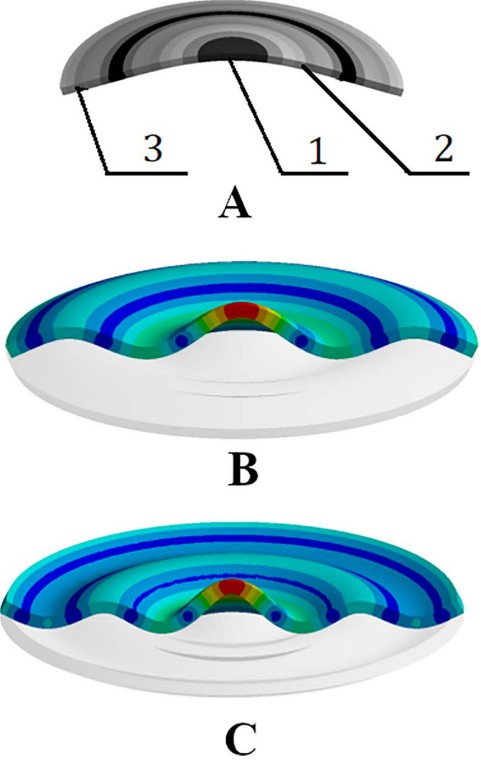

**Fig 2. Distribution of oscillation amplitudes of a flat disk.** (A) Second mode. (B) Third mode. (C) Fourth mode. 1 – central region; 2 – first ring zone; 3 – second ring zone.

to the lack of fixation along the outer edge. As a result, the oscillation amplitude at the periphery can increase by 30–70% compared to neighboring zones.

Preliminary calculations of the parameters of disk emitters are usually performed using the method described in works [9,23], and based on the dependence of the resonant frequency on the main geometric characteristics of the disk (thickness and diameter), according to expression (1):

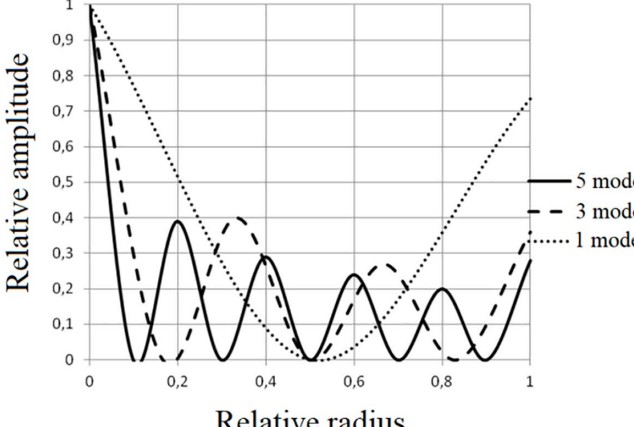

**Fig 3. Distributions of relative oscillation amplitudes along the radius of a flat disk oscillating in different modes.**

$$f = \frac{\pi h n^2}{d^2} \cdot \sqrt{\frac{E}{3\rho(1-\mu^2)}}$$

(1)

where $f$ is the resonant frequency of the emitter, Hz; $d$ is the emitter diameter, m; $h$ is the disk thickness, m; $E$ is Young's modulus, Pa; $\rho$ is the density, kg/m3; $\mu$ is Poisson's ratio, $n$ is the ring mode number. (For titanium VT-1–0 $E = 112 \cdot 10^9$; $\rho = 4500$ kg/m3; $\mu = 0.355$).

Increasing the disk diameter by using vibrations at higher modes (second and subsequent) leads to a decrease in the vibration amplitude in individual surface zones. Therefore, it is necessary to experimentally confirm the possibility of increasing the efficiency of emitters by increasing their diameter or to find alternative ways to solve the problem.

In this regard, the task was set to develop and study single-type ultrasonic emitters operating on modes, starting with the second. Theoretical analysis and modeling allowed us to select as experimental samples disks made of titanium alloys VT-5 and VT-6 with diameters of 99 and 146 mm. The thickness of the disks was selected in such a way as to ensure an operating frequency in the range of 20±2 kHz.

The ANSYS CFX finite element modeling system was used to calculate and analyze the vibration mode of an ultrasonic disk emitter. A model analysis was performed to determine the emitter's key frequency characteristics, using a tetrahedral finite element. During the modal analysis, a convergence analysis of the numerical results for various emitter designs was conducted. A modeling result was considered satisfactory if it corresponded to a finite element model with a minimum number of finite elements, an increase in which leads to a change in the key values of the design parameters (e.g., natural frequency of vibration) by no more than 0.2% to 0.5%.

The emitter with a titanium flat disk with a diameter of 99 mm, connected to a piezoelectric transducer, is shown in Fig 4A. The geometric parameters provide excitation of bending vibrations at a frequency corresponding to the resonance of the piezoelectric transducer. To increase the radiation area, an emitter with a flat disk with a diameter of 146 mm, of the corresponding thickness, was designed and implemented (Fig 4B) at the same frequency. Photos of the disks in Fig 4C.

The disk with a diameter of 146 mm, unlike the disk with a smaller diameter of 99 mm, is made with a thickening in the central part on the side of the connection with the piezoelectric transducer. The central thickened section has a height equal to the wavelength of ultrasonic vibrations in the gas, and a diameter corresponding to the length of the bending wave in the disk material. Additionally, in the center of the disk there is an annular groove with a diameter equal to half the length of the bending wave, and a depth of up to a quarter of the length of the longitudinal wave in the material. According

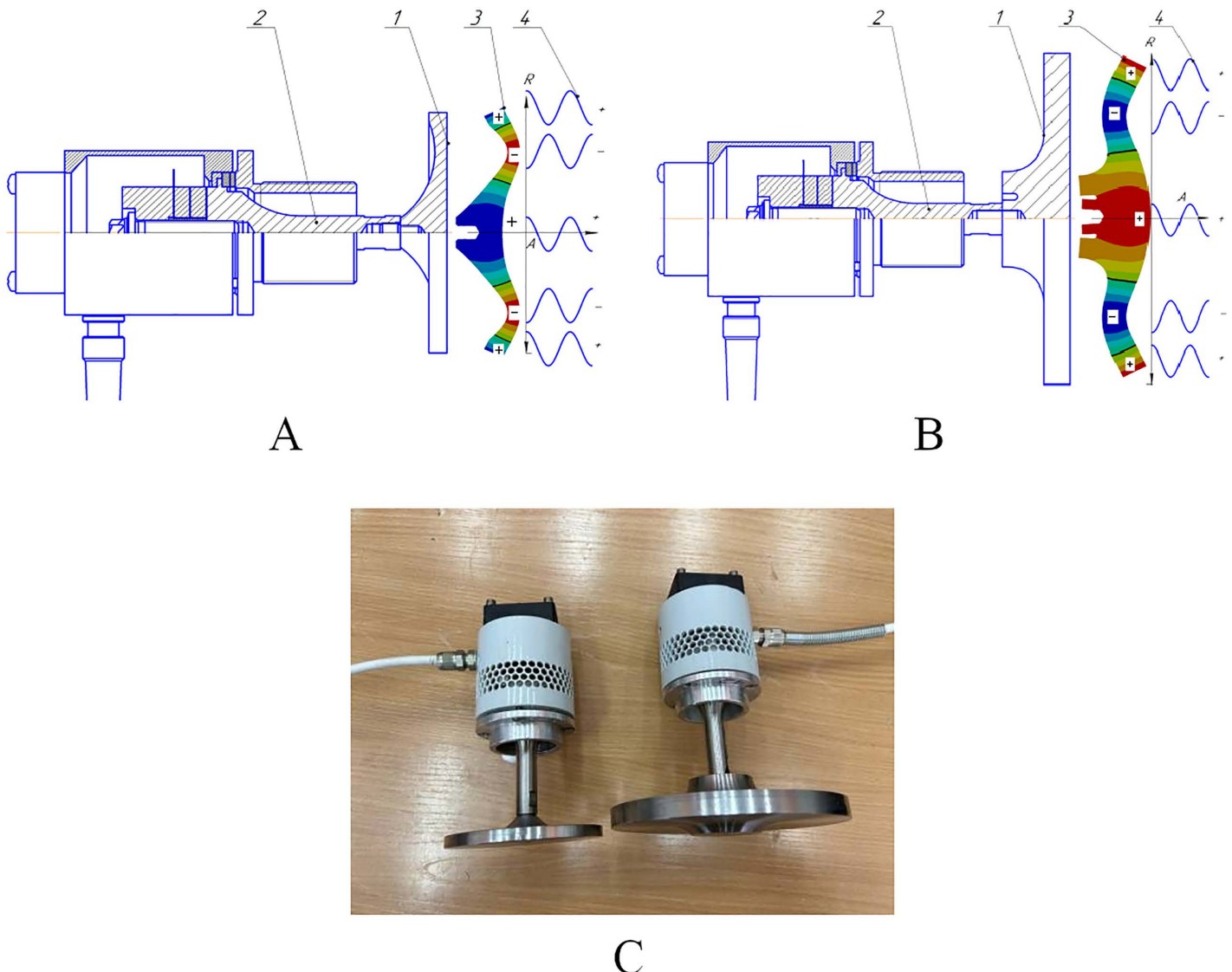

**Fig 4. Design diagrams, distribution of emitted vibrations and photos of emitters with disks of 99 and 146 mm diameter.** (A) Distribution for 99 mm disk. (B) Distribution for 146 mm disk. (C) Photos of emitters.

to the simulation results, such a design solution reduces mechanical stress in the connection zone and increases the efficiency of energy transfer from the transducer to the disk.

As can be seen from the oscillation distribution (Fig 4A and 4B), one of the key reasons for limiting the efficiency of emitters operating on higher modes is the presence of radiation zones on the disk surface oscillating in antiphase. Such zones emit oscillations that are mutually compensated at a certain distance from the radiating surface, reducing the total energy in the working volume of the gas. Tables 1 and 2 shows a comparison of the areas of the radiation zones of the two emitters under consideration.

Thus, increasing the diameter of the flexural-oscillating disk from 99 to 146 mm made it possible to increase the total radiation surface by 2.16 times. In this case, the area of the radiation surface oscillating with one phase (the central region

**Table 1. Comparison of modern ultrasonic emitters for gas environments.**

| Emitter type | Main design features | Operational frequency, kHz | SPL produced, dB | Efficiency, % | Advantages | Disadvantages |
|---|---|---|---|---|---|---|
| Gas jet (Hartman type and modifications) | Has a nozzle; gas jet excites the resonator | from 10 to 40 | from 120 to 130 | from 15 to 20 | Simplicity, autonomy | Low efficiency, large size |
| Langevin transducer | Has a piezo package, a concentrator and a tip | from 18 to 40 | from 135 to 145 | from 40 to 60 | Reliability, versatility | Bad wave impedance matching with air |
| Flat disk piezo-driven emitter | The piezoelectric element excites a metal disk | from 18 to 44 | from 135 to 147 | from 50 to 60 | Design simplicity, large radiating area | Antiphase zones, unevenness of radiation pattern |
| Step-profile piezo-driven disk emitter | Variable thickness for phase matching | from 18 to 44 | from 150 to 153 | from 60 to 65 | More uniform field, increased SPL | Complicated manufacturing |
| Piezo-driven emitter with phase-aligning horns | Disc or plate and reflectors | from 18 to 25 | 150–155 | up to 70 | Coherent radiation, using both sides of the disk | Large dimensions |
| Array of piezo-driven disk emitters | Several synchronized emitters | from 18 to 40 | >155 | from 60 to 70 | Controlled directivity pattern | Complexity of synchronization |

**Table 2. Radiation zones of oscillations with different phases.**

| Surface areas of disk zones | For a 99 mm disk | For a 146 mm disk |
|---|---|---|
| Total surface area of the disc, cm$^2$ | 77 | 167 |
| Central area, cm$^2$ | 23 | 29 |
| First ring zone, cm$^2$ | 37 | 96 |
| Second ring zone, cm$^2$ | 17 | 42 |
| Surface area with one phase, % | 52 | 43 |

and the second ring zone, as shown in Fig 2A) increased by 1.7 times. However, the first ring zone shown in Fig 2, oscillating in antiphase with respect to the central and second zones, also increased significantly. Therefore, the area of the surface oscillating in one phase decreased from 52% to 43% (by 1.2 times). The obtained result indicates that the use of disks with a diameter of less than 100 mm ensures predominant radiation of oscillations with one phase (more than 50%) due to the radiation of the central and extreme (second) zones of the disk, and the use of disks with a diameter of more than 100 mm ensures predominant radiation of oscillations with the opposite phase due to the radiation of the first ring zone of the disk (up to 57% for a disk with a diameter of 146 mm).

To increase the proportion of the surface participating in the in-phase radiation of oscillations, two methods are proposed.

1. Formation of a step-variable thickness of the disk, in which the height of each annular step compensates for the phase shift between adjacent zones (the emission of ultrasonic vibrations into the gas by all zones of the disk is carried out in phase).

2. Installation of horn devices in front of the antiphase oscillating sections of the disk. The dimensions of the horns are selected in such a way as to ensure phase matching of the ultrasonic oscillations at the horn output (a corresponding increase in the length of the path of ultrasonic oscillations from adjacent antiphase oscillating zones to the output of all horn devices, by analogy with a Fresnel lens).

The first approach is implemented by making a stepped disk, the design of which is shown in Fig 5.

The arrangement and dimensions of the steps determine the phase characteristics of the sound field formed by the emitter. Transitions between zones are performed in nodal circles at radii $D_1 \ldots D_{n-1}$ with a step height equal to half the ultrasound wavelength in gas. In this way, phase alignment is achieved over the entire surface and an in-phase plane wave is formed.

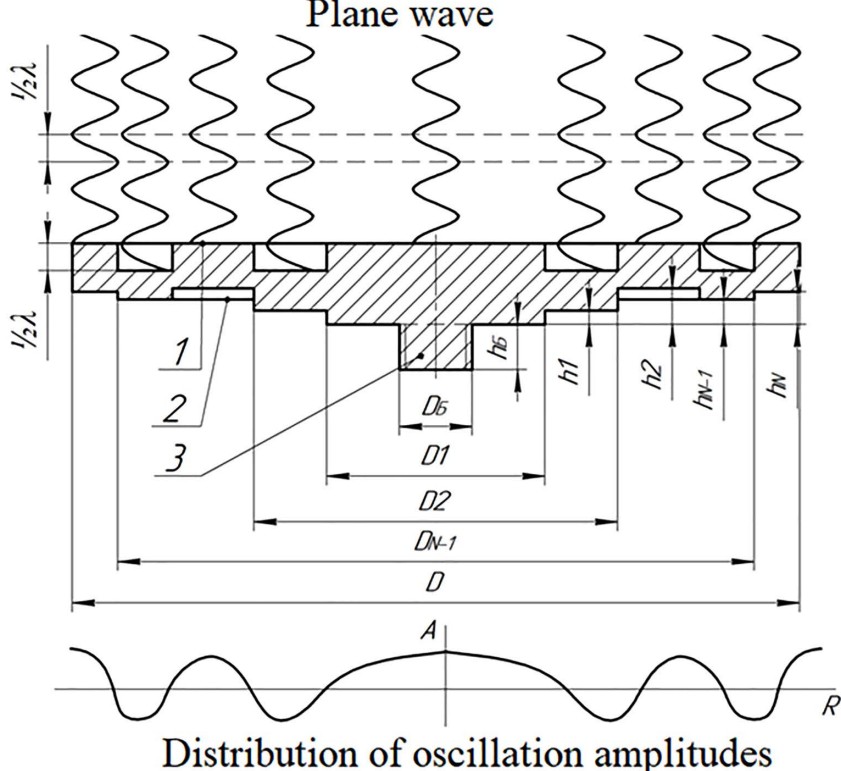

**Fig 5. Radiation from a disk with a step-variable surface.** 1 – front side; 2 – back side; 3 – connecting tailpiece.

In the design with a step-variable thickness, the increase in the rigidity of the annular zones along the radius is compensated, which allows the amplitude of oscillations to be leveled. The greatest efficiency is achieved when forming a relief (due to geometry) on the back side of the disk.

Below are presented the designs of ultrasonic emitters (Figs 6 and 7) for gas media with disks of different sizes, starting with a step-variable disk with a diameter of 146 mm, providing radiation in the second mode of the main oscillations of the disk (Fig 6).

This disk also has a variable thickness along the radius. The central section opposite the attachment zone with the transducer is thickened to a height equal to half the wavelength of the ultrasonic oscillations in the gas, and the outer zone is thinned to a value equal to the length of this wavelength. Analysis has shown that this solution allows increasing the proportion of the frontal surface participating in in-phase radiation from 43% to 75%.

Since a further increase in the effective radiation area is possible only with the use of higher oscillation modes (third, fourth, etc.), Fig 7 shows the designs of the created emitters equipped with step-profile disks with a diameter of 250, 320, 360 and 410 mm. All of them have a thickness variable in radius on the side of the connection with the piezoelectric transducer, which allows for maximum compensation of phase shifts between radiation zones and an increase in the efficiency of radiation into the gas environment.

## 2.2. Experimental setup

Specialized measuring stands were developed and used to conduct comparative measurements of the characteristics of ultrasonic emitters for gas environments. Measurements of energy characteristics consumed and emitted power, as well as the efficiency factor, were conducted on the stand described in [31,32].

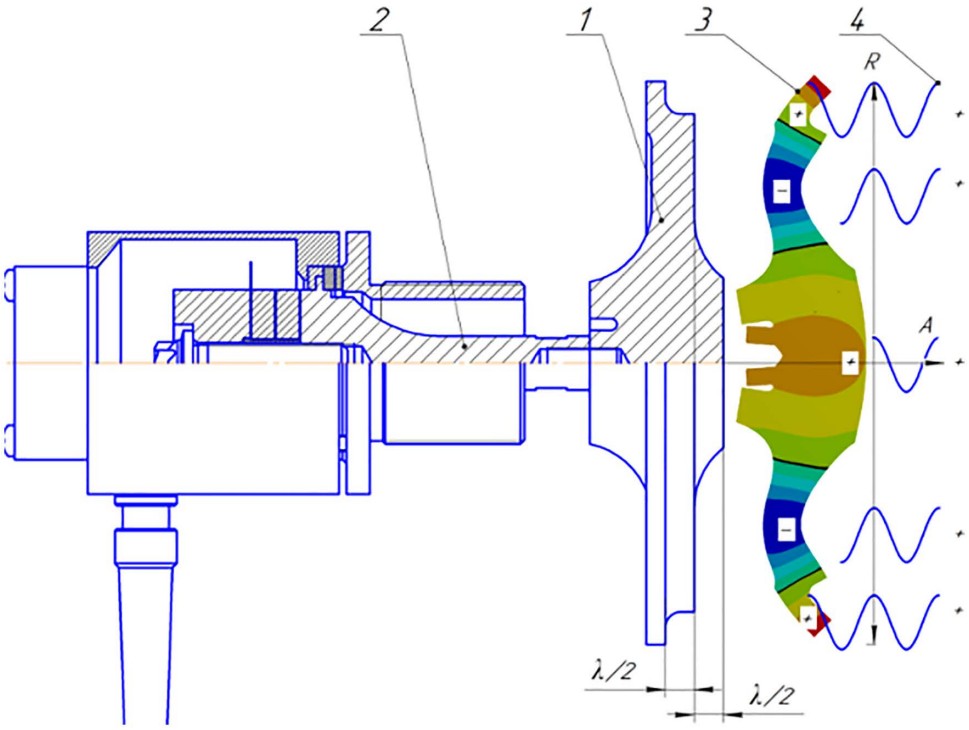

**Fig 6. Structural diagram and radiation of vibrations of a step-variable disk with a diameter of 146 mm.**

The setup enables comparative measurements to be performed when emitting ultrasonic vibrations into a vacuum chamber: at normal pressure and in the absence of air [32]. Air was evacuated from the vacuum chamber to a residual pressure not exceeding 1000 Pa. No vibrations were detected in the vacuum chamber (rarefied air environment), indicating the absence of acoustic radiation. Therefore, the electrical power consumed by the emitter in a vacuum can be estimated as the power of its own losses in the emitter material (i.e., without load – the emission of radiation into the gas). Thus, the efficiency of the ultrasonic emitter was determined as the ratio of the power of vibrations emitted into the air under normal conditions,

$$P_{ac.} = P_{full} - P_{loss},$$ (2)

where $P_{ac.}$ – acoustic power of the emitter; $P_{full}$ – total electrical power consumption of the emitter; $P_{loss}$ – the power of the emitter's own losses, to the total power consumption of the emitter when operating at normal pressure (3).

$$\eta = \frac{P_{ac.}}{P_{full}} 100\%,$$ (3)

where $\eta$ – efficiency coefficient of the emitter.

The electrical power consumed by the emitter was measured using a GPN-8212 meter. The errors in measuring the consumed and emitted power were determined by the errors of the standard measuring instruments used (GW Instek meter model GPN-8212).

To power the developed ultrasonic emitters, special electronic generators were used, which included both traditional solutions and new innovative developments [33–40].

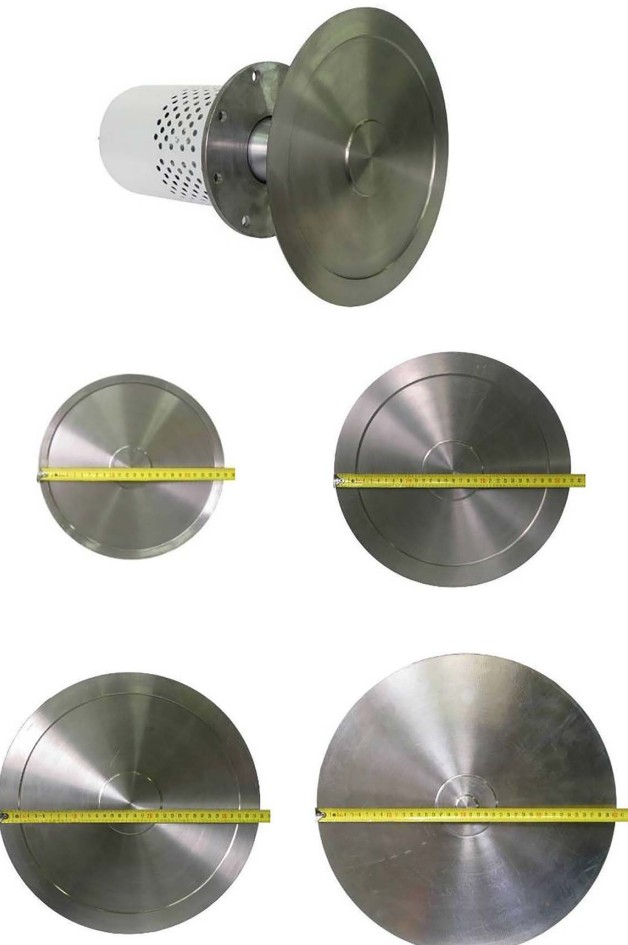

**Fig 7. Ultrasonic emitters with disks of 250, 320, 360 and 410 mm diameter.**

The stand for measuring acoustic radiation parameters – SPL, directivity patterns and sound pressure along the radiation axis up to 1 m – is shown in Fig 8.

During measurements, the ultrasound emitter was positioned vertically. The microphone (4) of the sound level meter (6) was located: at the center of the emitter; on the axis of symmetry of one or two ultrasound emitters; or at the center of the emitter array (depending on the type of emitter).

The ultrasonic transducer stand (2) was rigidly fixed. The microphone stand (5) was adjustable, changing the microphone's angle relative to the axis of symmetry. The angle varied from 0° to 180°. A distance of one meter was maintained between the radiating surfaces of the ultrasonic transducers and the microphone.

The errors in measuring the acoustic radiation parameters were determined by the errors of the used system for measuring high sound pressure levels Ecophysics-110A with the MK/VMK-401 microphone (sensitivity 1.6 mV/Pa, equivalent capacitance 6 pF), which allows measurements to be taken at levels up to 172 dB with an error of no more than 1% [41].

Each parameter was measured at least 4 times. The measurement results were processed, graphed, and analyzed using Microsoft Excel spreadsheets. Statistical evaluation methods for the results of multiple measurements were used, including averaging values and finding the standard deviation.

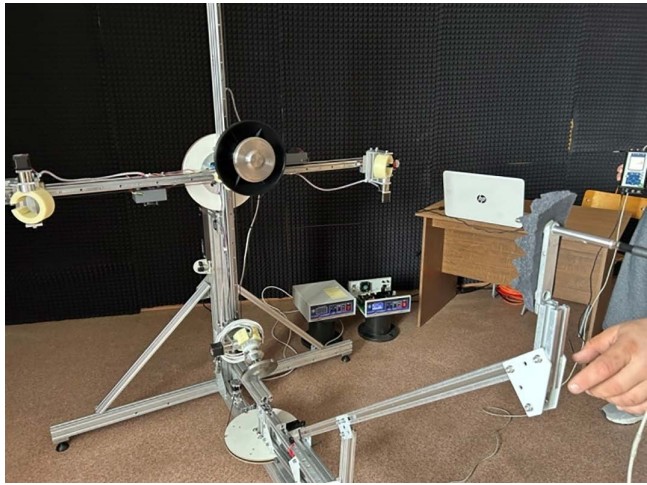

**Fig 8. Stand for measuring radiation parameters.**

## 3. Results and discussion

The technical characteristics of the developed ultrasonic emitters, measured using the developed stand, are given in Table 3.

At the first stage of the research, the directivity diagrams of two flat flexurally oscillating disks (without thickenings to compensate for phase differences in the oscillations of the ring sections) were obtained – with a diameter of 99 and 146 mm (Fig 9). The SPL at a distance of 1 m from the radiating surface are presented.

The analysis of the obtained data allows us to draw the following conclusions.

1. The maximum SPL for a disk with a diameter of 99 mm was 141.3 dB (232 Pa), while for a disk with a diameter of 146 mm it reached 147.5 dB (471 Pa).

2. An increase in the total radiation area by 2.16 times and an increase in the area oscillating in one phase by 1.7 times led to an increase in sound pressure by more than two times (by 6.2 dB).

At the second stage, two disks with a diameter of 146 mm were examined: one was flat, and the second was made stepwise variable to compensate for the phase differences in the oscillations of the ring sections (Fig 10).

The comparison showed that the maximum SPL of the flat disk was 147.5 dB (471 Pa), and of the stepped variable disk – 153.2 dB (914 Pa). Thus, the implementation of the 146 mm diameter disk stepped variable for phase alignment on the surface of the radiator provided an increase in sound pressure almost twofold (by 5.7 dB). The measurements carried

**Table 3. Technical characteristics of the developed ultrasonic emitters.**

| | Disc diameter, mm | | | | |
|---|---|---|---|---|---|
| **Characteristics** | 146 | 250 | 320 | 360 | 410 |
| Oscillation frequency, kHz | 22 ± 1,85 | 22,5 ± 1,85 | 30 ± 2,2 | 29 ± 2,3 | 27 ± 2,0 |
| Emitter's length, mm | 200 | 380 | 400 | 410 | 450 |
| Power consumption, VA | 180 ± 9 | 300 ± 15 | 350 ± 17 | 380 ± 19 | 450 ± 23 |
| Radiated acoustic power, VA | 115 ± 6 | 180 ± 9 | 210 ± 10 | 220 ± 11 | 225 ± 12 |
| Efficiency, % | 63 ± 3 | 60 ± 3 | 60 ± 4 | 58 ± 4 | 50 ± 5 |

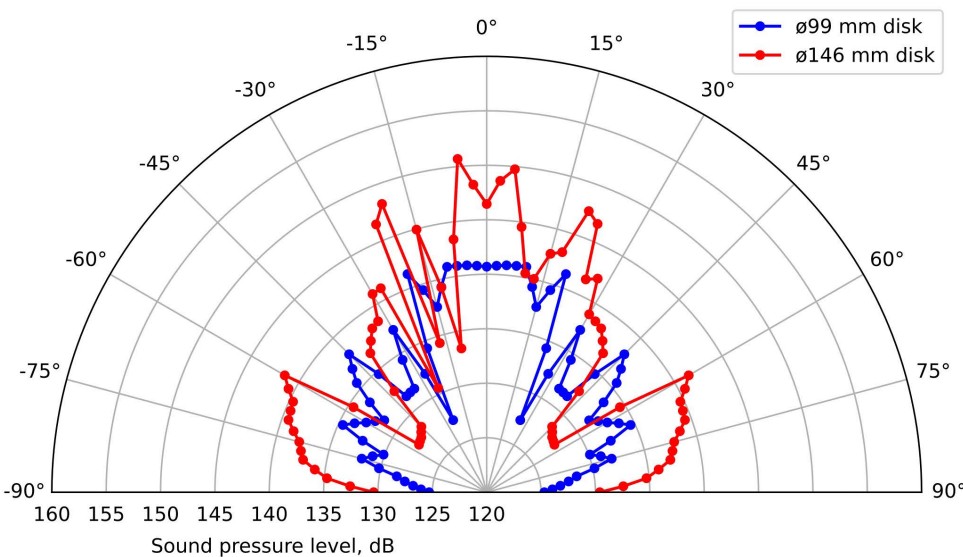

**Fig 9. Radiation patterns for two flat disks with a diameter of 99 mm (blue) and 146 mm (red).**

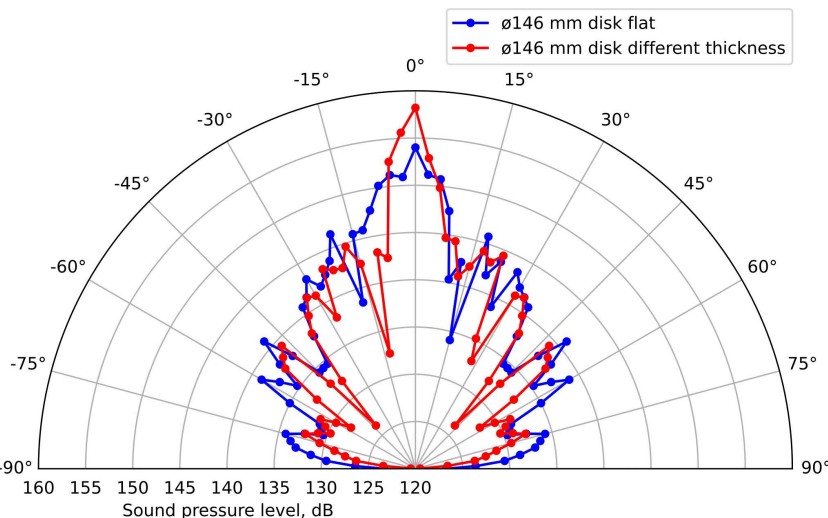

**Fig 10. Radiation patterns for two disks with diameters of 146 mm: flat (blue) and of different thicknesses (red).**

out confirm that the use of disks with stepped variable thickness allows for a significant increase in the radiation efficiency (proportionally to the increase in the surface area radiating oscillations in one phase) due to the in-phase operation of most of the surface.

The results of measuring the sound pressure distribution along the emitter axis up to a distance of 1 m are shown in Fig 11. The data show characteristic jump-like changes (in the near zone), especially pronounced for the stepped disk (marked in red), associated with the interference of oscillations from different annular zones oscillating both in the same and in opposite phases.

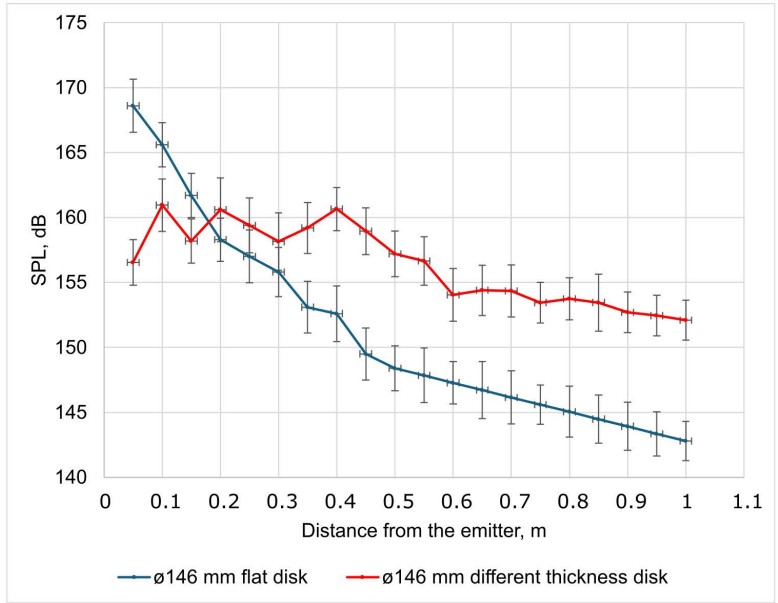

**Fig 11. Distribution of sound pressure along the emitter axis up to a distance of 1 m for flat (blue) and multi-thickness (red) disks.**

The zone of active interference of oscillations ends at a distance of about 0.6 m from the emitter, which corresponds to approximately four disk diameters. At the same time, a uniform decrease in sound pressure begins at a distance of 0.40 to 0.45 m (≈ 3 disk diameters). Thus, it can be considered that the stable formation of the acoustic field (far field) begins at a distance equal to 3–4 emitter diameters. This corresponds to theoretical ideas about the near and far fields of flat emitters. This is typical for all emitters considered below.

At the next stage, the radiation patterns of emitters with disks of diameters 250, 320, 360 and 410 mm, operating in the third, fourth, fifth and sixth oscillation modes, respectively, were obtained (Fig 12).

A comparative analysis showed the following: for a disc with a diameter of 250 mm, the SPL at a distance of 1 m was 140 dB (195 Pa); with a diameter of 320 mm – 143 dB (350 Pa); with a diameter of 360 mm – 148 dB (435 Pa); with a diameter of 410 mm – 150 dB (700 Pa).

Thus, successive increase of the disk diameter to 250 mm, 320 mm, 360 mm and 410 mm and implementation of stepwise variable thickness allows to increase the SPL from 140 dB (195 Pa) to 150 dB (700 Pa). However, the obtained values are inferior to the results achieved for a stepwise variable disk with a diameter of 146 mm, operating in the second mode. It was also found that an increase in the diameter leads to an expansion of the directivity pattern. This is partly explained by the fact that the measurements were carried out in the near zone, which does not allow the directional radiation to fully form (for a 410 mm disk, the near zone exceeds 1.2–2 m).

However, the main reason for the decrease in the efficiency of large-diameter disks, compared to a disk with a diameter of 146 mm, is the asymmetry of the generated oscillations. For clarification, Fig 13 shows an image of the surface of a disk with a diameter of 410 mm with indicator powder applied. It is evident that in the area of intersection of diagonals, the nodal lines are curved, which indicates distortions of oscillations and a decrease in amplitudes in individual ring zones.

The reason for such distortions is related to the anisotropy of the sheet metal used to manufacture large disks. Unlike cylindrical blanks (e.g., 99 and 146 mm diameters), sheet metal exhibits different mechanical properties along and across the rolling direction.

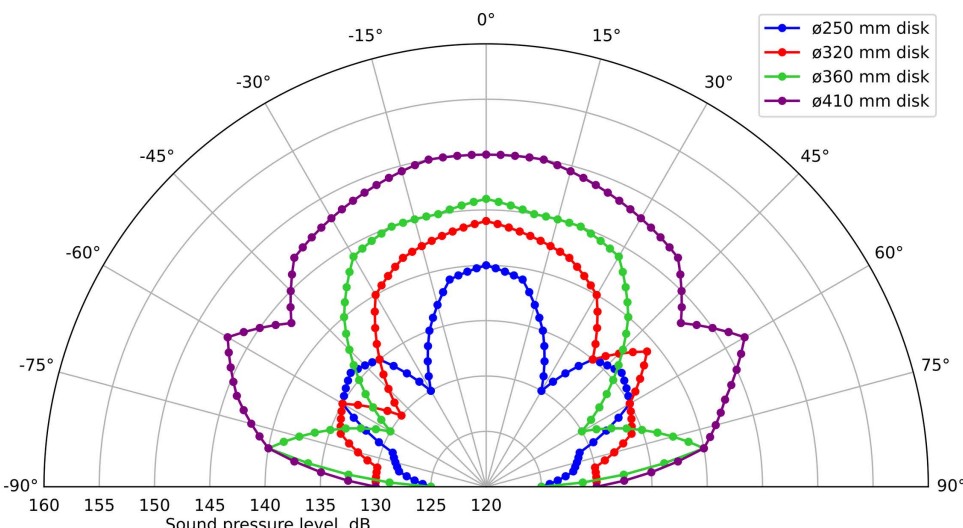

**Fig 12. Directional patterns of ultrasonic emitters with disks of diameters 250, 320, 360 and 410 mm.**

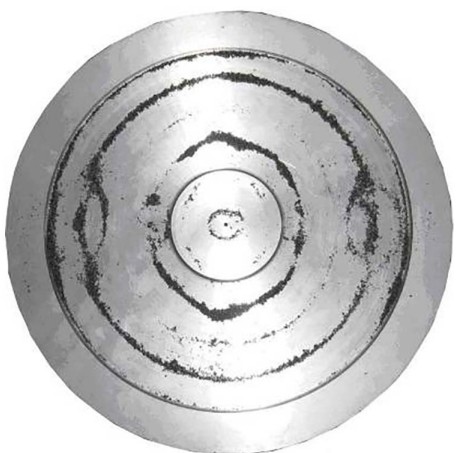

**Fig 13. Zero-oscillation lines of a disk emitter with a diameter of 410 mm.**

Based on the data obtained, it can be concluded that increasing the disk diameter is effective only to a certain limit – in particular, to the possibility of operating on the second oscillation mode. Further increasing the size and using higher modes leads to a slowdown in the growth of efficiency, accompanied by significant costs for materials, manufacturing and energy supply.

Therefore, the next stage of the work was aimed at implementing the second approach to increasing the radiation efficiency – the use of phase-correcting horn devices. To increase the sound pressure of a disk with a diameter of 99 mm, a solution was implemented to compensate for phase shifts occurring between the radiation zones. On the side opposite the connection with the piezoelectric transducer, ring horns were installed, located at a distance of less than a quarter of the ultrasound wavelength in the gas (Fig 14).

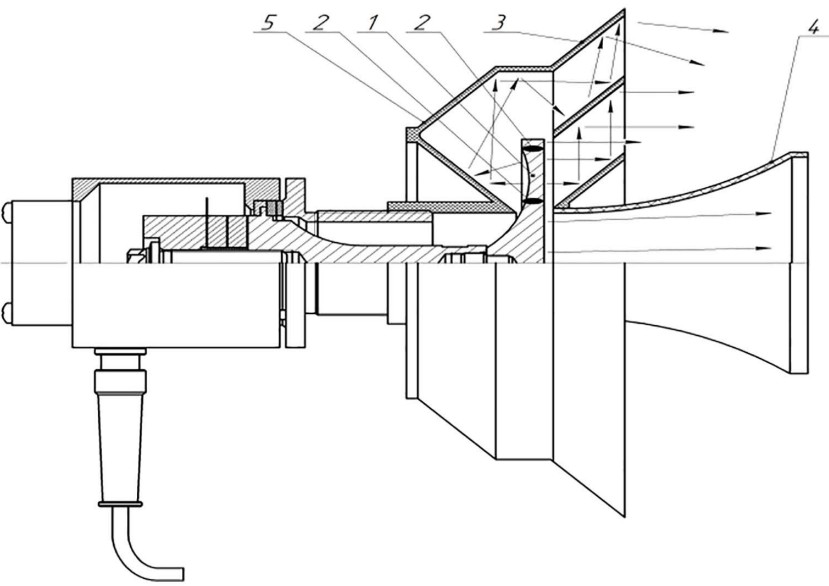

**Fig 14. Ultrasonic emitter with a 99 mm diameter disk with phase-aligning front horns and an oscillation reflector.**

The horns are made in the form of annular truncated cones with an opening angle of 90°. The central (first) and all odd horns are located above the zones oscillating in phase with the converter. The diameter of each cone increases from the diameter of the zone to the maximum diameter of the corresponding annular region. Even horns installed above the zones with the opposite phase have an initial diameter increased by half the wavelength of ultrasound in the gas, providing phase compensation when exiting the horn. On the side of the disk mount, there is an additional reflector made in the form of two concentric truncated cones with a similar opening angle, connected to each other and attached to the converter body.

The next stage of testing was carried out with emitters based on disks with a diameter of 146 mm (Fig 15). One of them was equipped with front phase-equalizing horns (for a flat disk), and the second (in the photo) – without them, but with a profiled step-variable surface. In both cases, rear reflectors were used, taking into account the need to compensate for the phases of oscillations of the back side of the disk.

The results of measuring SPL at a distance of 1 m from the emitter for three types of discs (flat, with diameters of 99 and 146 mm, and of different thicknesses with a diameter of 146 mm) are presented in Fig 16.

Analysis of the obtained data shows that the use of front phase-equalizing and rear reflective horns allows for a significant increase in radiation efficiency. In particular, the maximum SPL for a flat disk with a diameter of 99 mm increased from 141.3 dB (232 Pa) to 148.2 dB, i.e., by 6.9 dB, while the directivity pattern (the width of the main lobe) remained virtually unchanged.

For a flat disc with a diameter of 146 mm, the sound pressure increased from 147.5 dB (471 Pa) to 155 dB, i.e., by 7.5 dB. A disc of the same diameter with a stepped thickness (different thickness) ensured the maximum SPL of 159.2 dB, which is more than twice the initial value (in terms of sound pressure in Pascals). Thus, using only a rear reflector for this disc ensured an increase in sound pressure from 153.2 dB (914 Pa) to 159.2 dB (1824 Pa), i.e., by 6 dB.

From these results it follows that the use of phase-correcting and reflective devices makes it possible to achieve a significant increase in the SPL, which cannot be achieved by simply increasing the diameter of the disks and moving to higher vibration modes.

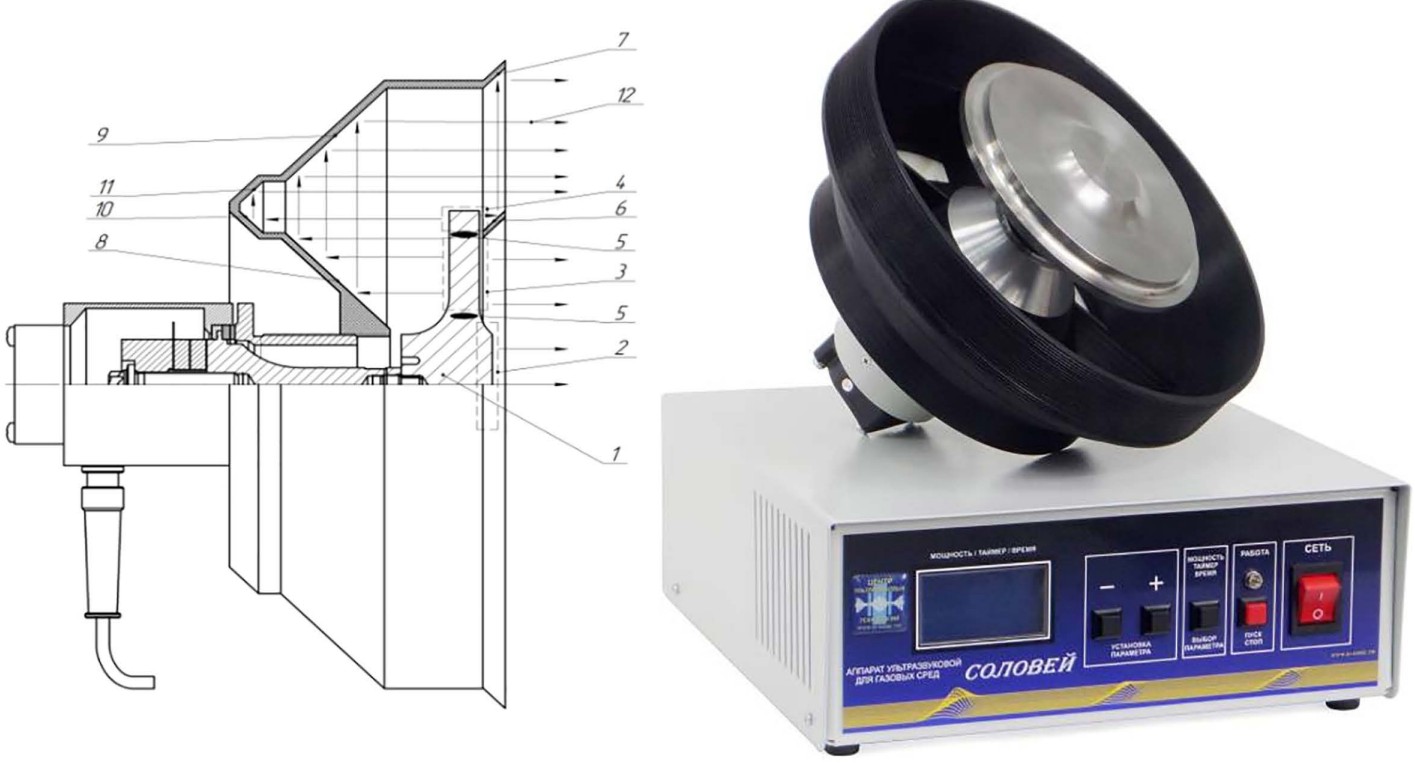

**Fig 15. Ultrasonic emitter with phase-aligning horns.**

## 4. Conclusions

The conducted studies confirmed the high efficiency of the developed disk radiators operating in the second bending mode and demonstrated the limitations of the traditional approach based on increasing the diameter and transitioning to higher modes. Although stepped-profile disks with diameters from 250 to 410 mm provided an SPL increase from 140 to 150 dB, these values are inferior to the results achieved with a disk with a diameter of 146 mm.

The main advantage of the developed solutions is the use of a stepped-profile surface, which compensates for phase shifts between the annular zones and ensures in-phase radiation across the entire disk area. This increased the proportion of the surface operating in phase from 43% (for a 146 mm flat disk) to 75% and increased the sound pressure level from 147.5 dB (471 Pa) to 153.2 dB (914 Pa).

Further use of phase-aligning horns and rear reflectors provided further efficiency gains. For a flat disc with a diameter of 146 mm, the SPL increased from 147.5 dB to 155 dB, and for a stepped disc of the same diameter, from 153.2 dB to 159.2 dB (1824 Pa). A similar approach for a disc with a diameter of 99 mm increased the SPL from 141.3 dB (232 Pa) to 148.2 dB, confirming the versatility of the method.

Thus, it has been established that:

- it is advisable to create radiators operating in the second mode of bending vibrations;

- step-profile disk geometry is an effective solution for phase matching of radiating zones;

- the use of phase-aligning horns and reflectors allows sound pressure levels up to 159.2 dB to be achieved, significantly exceeding those of known designs.

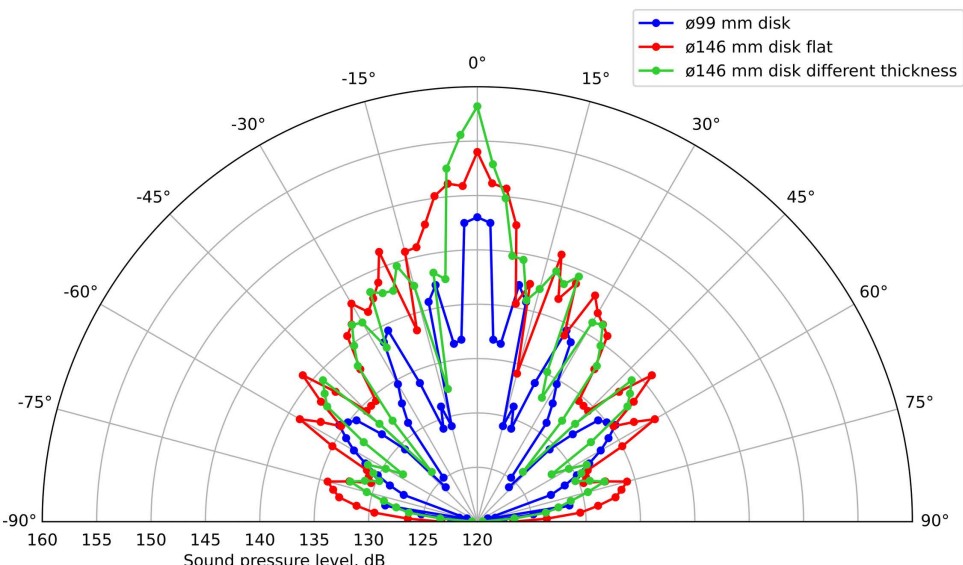

**Fig 16. Directional patterns for a 99 mm diameter disc (blue) and 146 mm diameter discs (flat – red and of different thicknesses – green) with horns.**

Future research opportunities include the development of multi-disk radiating arrays based on 99–146 mm diameter radiators, optimization of horn profiles for various frequency ranges, and the implementation of active phase control systems. This will expand the radiators' functionality, enable the generation of acoustic fields with a given directionality, and improve their efficiency in industrial, environmental, and security applications.

This work was supported of the grant of the Russian Science Foundation No. 24-19-00900, https://rscf.ru/project/24-19-00900/ (accessed on April 6th, 2025).

## Appendix A

List of abbreviations and symbols used

US – ultrasonic;

dB – decibels;

SPL – sound pressure level;

$f$ – the resonant frequency of the emitter, Hz;

$d$ – the emitter diameter, m;

$h$ – the disk thickness, m;

$E$ – Young's modulus, Pa;

$\rho$ – the density, kg/m3;

$\mu$ – Poisson's ratio;

$n$ – the ring mode number;

$P_{ac.}$ – acoustic power of the ultrasonic emitter;

$P_{full}$ – total electrical power consumption of the emitter;

$P_{loss}$ – emitter's power loss;

$\eta$ – emitter's efficiency.

## Supporting information

**S1 Table. Figure 9 initial data.** XLSX table containing data for plotting Fig 9. (XLSX)

**S2 Table. Figure 10 initial data.** XLSX table containing data for plotting Fig 10. (XLSX)

**S3 Table. Figure 11 initial data.** XLSX table containing data for plotting Fig 11. (XLSX)

**S4 Table. Figure 12 initial data.** XLSX table containing data for plotting Fig 12. (XLSX)

**S5 Table. Figure 16 initial data.** XLSX table containing data for plotting Fig 16. (XLSX)

## Author contributions

**Conceptualization:** Vladimir Khmelev.

**Data curation:** Sergey Tsyganok.

**Formal analysis:** Andrey Shalunov, Pavel Danilov.

**Funding acquisition:** Andrey Shalunov.

**Methodology:** Vladimir Khmelev.

**Project administration:** Andrey Shalunov.

**Resources:** Sergey Tsyganok, Pavel Danilov.

**Visualization:** Sergey Tsyganok, Alexander Sinkin.

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
