## [Decision Letter · Decision Letter 0]

4 Sep 2025

Dear Dr. Shalunov,

The manuscript needs more in-depth explanation and better clarity for figures and text.

We look forward to receiving your revised manuscript.

Kind regards,

Massimo Mariello

Academic Editor

PLOS ONE

Additional Editor Comments (if provided):

Reviewer #1:

Reviewer #2:

Reviewer #3:

Reviewers' comments:

Reviewer's Responses to Questions

**Comments to the Author**

1. Is the manuscript technically sound, and do the data support the conclusions?

Reviewer #1: Yes

Reviewer #2: Yes

Reviewer #3: Partly

2. Has the statistical analysis been performed appropriately and rigorously?

Reviewer #1: No

Reviewer #2: Yes

Reviewer #3: Yes

3. Have the authors made all data underlying the findings in their manuscript fully available?

Reviewer #1: No

Reviewer #2: Yes

Reviewer #3: Yes

4. Is the manuscript presented in an intelligible fashion and written in standard English?

Reviewer #1: No

Reviewer #2: Yes

Reviewer #3: Yes

Reviewer #1: Please find my comments below:

1. The manuscript should be structured properly, and headings should be numbered.

2. It is suggested to cite more recent literature (2020–2025), especially from outside the authors' own group.

3. In the introduction section, provide the background of the topic, and motivation and objectives of this work. Summarize the related work in literature and narrow down to the research gap. Also include a comparative table for modern ultrasonic emitters.

4. The authors have reported SPL increases, however, uncertainty ranges, error bars, and number of repetitions are missing in the results.

5. The manuscript has several grammatical errors. Please consider thorough language editing.

6. Please include a nomenclature to define all symbols along with their units.

7. Please ensure that all the data is provided with the supporting files.

Reviewer #2: The paper presents new approaches to increasing sound pressure during oscillation generation in gas media of ultrasonic flexural-mode emitters. The work is interesting and the paper well written, but the following issues should be carefully addressed [(R) and (S) is defined at the end of the comments]:

1. (R) Abstract. The abstract also should present a brief topic framework, describe the scope and objectives, in addition to present the methodology adopted ind the main results/conclusions.

2. (R) Page 8, paragraph 2. Replace “wave,and” with “wave, and”.

3. (R) Page 10. Paragraph 1 after Fig. 5. variables should be italicised.

4. (R) Table 2. Overall dimensions, mm: disc diameter d x D. Could authors clarify the meaning of "x D"?, For example, in disc diameter 146 mm, it is 146x200.

5. (S) Figures 9, 10, 12, 16. All graphs could have the same scale to make it easier to compare the results. For example, 110, 115, 120,... 160.

6. (R) Figures 9, 10, 12, 16. The units should be shown.

(S) Suggestion. The author can change it or not. Does not compromise scientific rigour or understanding of the document.

(R) Recommendation. It does not compromise scientific rigour, but it compromises the rigour of communication and understanding.

Reviewer #3: The reviewer appreciates the work done by the authors. The contents of the article are generally interesting and promising for further practical applications. The manuscript is basically properly structured. It is suitable for publication in the “Plos One (Public Library of Science)”. Anyway, the goal of the work must be better illustrated within the abstract, introduction and conclusions. Compared with other research articles, which treat the same topic, the superiority of the manuscript has not been clearly explained. So, please review the corresponding parts. Moreover, please make sure that all the software, which have been utilized, have been referenced. The further work should additionally be mentioned at the end of the article. Furthermore, some sentences within the text are too long. So, spelling, sentence structure, conciseness and writing style could be improved with the help of a native English speaker. In conclusion, an “appendix” section, containing names and elaboration of the symbols and acronyms used, should also be inserted at the end of the article.

**Do you want your identity to be public for this peer review?** For information about this choice, including consent withdrawal, please see our Privacy Policy

Reviewer #1: No

Reviewer #2: No

Reviewer #3: No

---

## [Author Response · Author response to Decision Letter 1]

8 Oct 2025

Reviewer 1

Thank you very much for your valuable suggestions regarding our work. Below we have tried to provide answers to all questions. Appropriate corrections have been made to the article. We hope that after the correction the article became significantly better.

1. The manuscript should be structured properly, and headings should be numbered.

Thank you for your comment. We have made every effort to align the manuscript's structure with PLOS ONE journal requirements. We have numbered the headings. The corresponding corrections have been made to the text.

2. It is suggested to cite more recent literature (2020–2025), especially from outside the authors' own group.

Thank you for your comment. The corresponding corrections have been made to the list of references. Currently at least 38% of the total number are sources from 2020-2025.

3. In the introduction section, provide the background of the topic, and motivation and objectives of this work. Summarize the related work in literature and narrow down to the research gap. Also include a comparative table for modern ultrasonic emitters.

Thank you. We completely agree with you. We have completely rewritten the introduction to provide basic information about the research topic and the current state of the art. We also outline the current research challenges and our proposed solutions. A comparative table of ultrasonic emitter characteristics has been added at the end of the introduction (Table 1).

4. The authors have reported SPL increases, however, uncertainty ranges, error bars, and number of repetitions are missing in the results.

Thank you for noticing this unfortunate oversight. We've updated the article with more detailed information about the experimental setup (page 14), uncertainty ranges for the SPL values (page 15), and information on the number of repeated measurements (page 15).

Unfortunately, adding error bars to the graphs overcrowds them with overlapping bars and thus makes them unreadable. We have added the description of the error ranges regarding measurements in the text of the article (see page 15)

5. The manuscript has several grammatical errors. Please consider thorough language editing.

Thank you for bringing this error to our attention. We've reviewed the article and attempted to correct any spelling errors.

6. Pleaseincludeanomenclatureto define all symbols along with their units.

Thank you for your comment. A list of all symbolic notations and units of measurement has been added to the article as Appendix 1.

7. Please ensure that all the data is provided with the supporting files.

Thank you for your comment. We've attached the data needed to construct the radiation patterns to the article.

Reviewer 2:

Thank you very much for your time and appreciation of our manuscript. We tried to take into account all your suggestions and correct your comments. We hope that after correction the manuscript has become significantly better.

1. Abstract. The abstract also should present a brief topic framework, describe the scope and objectives, in addition to present the methodology adopted ind the main results/conclusions.

Thank you for your work. The abstract has now been updated to include the purpose of the research, allowing for a clearer understanding of the article's focus. The abstract presents the main results achieved during the study in more detail. Potential applications of the research findings are outlined, particularly the generation of directional acoustic fields for solving a wide range of technological problems.

2. Page 8, paragraph 2. Replace “wave,and” with “wave, and”.

Thank you for noticing this unfortunate oversight. We'vecorrectedit.

3. Page 10. Paragraph 1 after Fig. 5. variables should be italicised.

Thank you. After Figure 5, the variables are highlighted in italics.

4. Table 2. Overall dimensions, mm: disc diameter d x D. Could authors clarify the meaning of "x D"?, For example, in disc diameter 146 mm, it is 146x200.

Thank you for your comment. The second 200 value (in the example provided), is the length of the emitter assembly with the piezoelectric transducer (the linear dimension along the centerline of the disk emitter). Since the disk diameter is known from the table column headers, we have changed the table format. This row now only indicates the emitter length.

5. Figures 9, 10, 12, 16. All graphs could have the same scale to make it easier to compare the results. For example, 110, 115, 120,... 160.

Thank you for your attention. The graphs in Figures 9, 10, 12, and 16 are shown to the same scale.

6. Figures 9, 10, 12, 16. The units should be shown.

Thank you. We have indicated the units of measurement in figures 9, 10, 12, and 16.

Reviewer №3:

Thank you very much for the work done and the critical comments provided on the article in question.

Anyway, the goal of the work must be better illustrated within the abstract, introduction and conclusions. Compared with other research articles, which treat the same topic, the superiority of the manuscript has not been clearly explained. So, please review the corresponding parts. Moreover, please make sure that all the software, which have been utilized, have been referenced. The further work should additionally be mentioned at the end of the article. Furthermore, some sentences within the text are too long. So, spelling, sentence structure, conciseness and writing style could be improved with the help of a native English speaker. In conclusion, an “appendix” section, containing names and elaboration of the symbols and acronyms used, should also be inserted at the end of the article.

Thank you. We have added the abstract which describes the purpose of the research in more detail (p. 2). The introduction presents a breakdown of the stated goal for achieving it through solving the stated objectives (p. 3).

We carefully read and edited the text with the help of a native English speaker, making every effort to remove long sentences and correct spelling.

Links to the software used in the research are provided on page 15.

The Conclusions section (pp. 21, 22) presents the solved objectives, providing qualitative and quantitative data for each. The conclusion also highlights future prospects that arise from continuing the research topic.

The article is supplemented with an Appendix A containing a detailed description of the symbols and abbreviations used.

---

## [Decision Letter · Decision Letter 1]

30 Oct 2025

Ultrasonic flexural mode emitters: new approaches to increasing sound pressure during oscillation generation in gas media

PONE-D-25-39068R1

Dear Dr. Shalunov,

We’re pleased to inform you that your manuscript has been judged scientifically suitable for publication and will be formally accepted for publication once it meets all outstanding technical requirements.

Kind regards,

Massimo Mariello

Academic Editor

PLOS ONE

Additional Editor Comments (optional):

Reviewers' comments:

Reviewer's Responses to Questions

**Comments to the Author**

Reviewer #1: All comments have been addressed

Reviewer #2: All comments have been addressed

Reviewer #3: All comments have been addressed

Reviewer #4: All comments have been addressed

Reviewer #5: All comments have been addressed

Reviewer #6: (No Response)

Reviewer #7: All comments have been addressed

2. Is the manuscript technically sound, and do the data support the conclusions?

Reviewer #1: Yes

Reviewer #2: Yes

Reviewer #3: Yes

Reviewer #4: Yes

Reviewer #5: Yes

Reviewer #6: Partly

Reviewer #7: Yes

3. Has the statistical analysis been performed appropriately and rigorously?

Reviewer #1: Yes

Reviewer #2: Yes

Reviewer #3: Yes

Reviewer #4: N/A

Reviewer #5: N/A

Reviewer #6: Yes

Reviewer #7: Yes

4. Have the authors made all data underlying the findings in their manuscript fully available?

Reviewer #1: Yes

Reviewer #2: Yes

Reviewer #3: Yes

Reviewer #4: Yes

Reviewer #5: Yes

Reviewer #6: Yes

Reviewer #7: Yes

5. Is the manuscript presented in an intelligible fashion and written in standard English?

Reviewer #1: Yes

Reviewer #2: Yes

Reviewer #3: Yes

Reviewer #4: Yes

Reviewer #5: Yes

Reviewer #6: No

Reviewer #7: Yes

Reviewer #1: (No Response)

Reviewer #2: The paper presents new approaches to increasing sound pressure during oscillation generation in gas media of ultrasonic flexural-mode emitters. The work is interesting and the paper well written. The authors present an excellent work.

Reviewer #3: The required revisions have been carried out and the manuscript can be accepted for publication in the "Plos One" .

Reviewer #4: The paper is revised accordingly and can be accepted for publication. The authors have revised the paper base on my comments.

Reviewer #5: Dear Authors, I have reviewed the revised version of your manuscript and find it to be substantially improved compared to the earlier submission. You have clearly and satisfactorily addressed all of the reviewers comments.

Reviewer #6: Title: Ultrasonic flexural mode emitters: new approaches to increasing sound pressure during oscillation generation in gas media.

General comments: The authors have focused on the development and study of ultrasonic flexural-oscillating disk emitters for gas environments, generating elastic vibrations at ultrasonic frequencies (above 20 kHz) with high sound pressure levels required for energy-intensive technological processes (sound pressure levels exceeding 140 dB). The aim of the study was to identify the limitations of traditional flat disk designs and to substantiate new technical solutions that can significantly improve radiation efficiency in gas environments.

The article is well organized, fits the journal scope, and has a contribution. In addition, the authors tried to answer all reviewer questions. However, the revision should take into account the following points:

1. The authors should point out the main contribution (in the introduction and abstract sections) of their work. What is new about it?

2. The figures and tables in the manuscript are sufficient, but several figures require enhanced resolution and consistent formatting to improve clarity. Refining resolution, adjusting labels, or optimizing layout could make them more effective and readable. In addition, all figures should be the same size.

3. The authors should add physical explanations for the discussions.

4. Some advanced concepts should be referred to with adequate references for less experienced readers.

5. The authors should state the limitations of their study, theory, methods, or argument. These should be recognized and explicitly addressed to facilitate a more equitable and transparent assessment of their work. Identifying potential constraints or uncertainties would enhance the credibility and robustness of their conclusions.

6. There are many errors, so the authors need to check the grammar, typos, and errors in the manuscript.

In general, I do not recommend the article in its current form. If the author gives a convincing answer to the above items and discusses the innovation of the article, then a decision can be made about the article.

Reviewer #7: All the queries by other reviewer's have been answered adequately by the authors. The MS can be recommended for its publication.

**Do you want your identity to be public for this peer review?** For information about this choice, including consent withdrawal, please see our Privacy Policy

Reviewer #1: No

Reviewer #2: No

Reviewer #3: No

Reviewer #4: No

Reviewer #5: No

Reviewer #6: No

Reviewer #7: **Yes: ** Mriganka Shekhar Chaki

---

## [Editor Report · Acceptance letter]

PONE-D-25-39068R1

PLOS ONE

Dear Dr. Shalunov,

I'm pleased to inform you that your manuscript has been deemed suitable for publication in PLOS ONE. Congratulations! Your manuscript is now being handed over to our production team.

Kind regards,

on behalf of

Dr. Massimo Mariello

Academic Editor

PLOS ONE